# Quantitative evaluation of choriocapillaris using optical coherence tomography and optical coherence tomography angiography in patients with central serous chorioretinopathy after half-dose photodynamic therapy

**Hyun Seung Yang[1], Tae Gu Kang[2], Hyun Park[3], Ji Su Heo[1], Jonghoon Park[1], Kyung Sub Lee[1], Sangkyung Choi[2]***

1 Department of Ophthalmology, Seoul Shinsegae Eye Center, Eui Jung Bu, Gyeonggi-do, South Korea,
2 Department of Ophthalmology, Veterans Health Service Medical Center, Seoul, South Korea,
3 Department of Endocrinology, Seoul Chuk Hospital, Eui Jung Bu, Gyeonggi-do, South Korea

* pfskchoi@gmail.com

## Abstract

### Purpose

To quantify the structural and perfusion changes in choriocapillaris in chronic central serous chorioretinopathy after half-dose photodynamic therapy by using spectral-domain optical coherence tomography and optical coherence tomography angiography.

### Methods

This retrospective interventional case series examined the eyes of patients with central serous chorioretinopathy. Patients underwent full ophthalmic examinations, including spectral-domain optical coherence tomography and angiography, prior to and 1, 3, and 6 months after the treatment. Clinical and tomographic features of the choriocapillaris and choroidal thickness and vascular changes were evaluated by assessing flow signal voids.

### Results

All 56 eyes of 56 patients showed complete resolution of subretinal fluid at 3 months after photodynamic therapy. The best-corrected visual acuity significantly improved at 6 months (p<0.001). The central subfield thickness, subfoveal choroidal thickness, subfoveal choroidal large vessel layer thickness, and mean total area of flow signal voids decreased significantly at 6 months (all p values < 0.001), but the subfoveal choriocapillaris layer thickness did not change significantly at 6 months (p≥0.16). Multivariate analysis revealed positive linear correlations of the central subfield thickness and subfoveal choroidal large vessel layer thickness with the mean total area of flow signal voids at 6 months (p<0.001). There was a negative linear correlation between the subfoveal choriocapillaris layer and the mean total area of flow signal voids at 6 months (p = 0.013).

**Data Availability Statement:** All relevant data are within the paper and its Supporting Information files.

**Funding:** This study was supported by a Veterans Health Service Medical Center Research Grant, Republic of Korea (grant VHSMC19027). The funder had no role in study design, data collection and analysis, decision to publish, or preparation of the manuscript.

**Competing interests:** The authors have declared that no competing interests exist.

## Conclusion

Half-dose photodynamic therapy improved the anatomic and functional outcomes of central serous chorioretinopathy, induced subfoveal choroidal thickness thinning, and increased choriocapillaris perfusion. In addition, the recovery of the subfoveal choriocapillaris layer thickness and improved choriocapillaris perfusion were closely associated.

## Introduction

Central serous chorioretinopathy (CSC) is a well-known vision-threatening disease that occurs in relatively young patients [1,2]. It is characterized by localized serous detachment of the retina that mainly involves the macular area, and predominantly affects males. CSC often resolves spontaneously, thus it usually has a favorable prognosis. However, untreated chronic CSC can result in severe retinal pigment epitheliopathy and severe vision loss.

Although there are many treatment options for CSC, including steroid discontinuation [3], anti-VEGF therapy [4], focal laser treatment [5], subthreshold micropulse laser [6–8], mineralocorticoid receptor antagonist therapy [9] and diuretics, photodynamic therapy (PDT) is believed to be the most effective and permanent treatment [10]. PDT with verteporfin induces the resorption of subretinal fluid (SRF) by reducing choroidal vascular hyperpermeability [5,10,11]. However, the mechanism underlying these effects remains unclear.

Recent advances in optical coherence tomography (OCT) and optical coherence tomography angiography (OCTA) allow noninvasive, high-resolution imaging of the choroidal structure and vasculature in many chorioretinal diseases. Previous studies have used these new imaging modalities to examine CSC patients treated with PDT [12,13]. Izumi et al. demonstrated using OCT that half-dose PDT reduces subfoveal choroidal thickness (SFCT) by decreasing the large choroidal vessel layer but without altering the choriocapillaris-medium choroidal layer [12,14,15]. Demircan et al. and Nassisi et al. used OCTA to show that the thickness of the choriocapillaris appeared to decrease in the early stages of chorioretinal diseases and recovered to baseline or even higher levels thereafter [13,14]. Demirel et al. also reported a significant increase in choriocapillaris perfusion in CSC patients after half-fluence PDT [15]. However, the image quality in OCTA varies depending on the signal intensity because of media opacities and SRF. To minimize the impact of these variations, Spaide et al. suggested a reproducible quantitative method that aims to measure the dark areas representing absent or decreased flow signal within the choriocapillaris, also known as flow signal voids [16]. However, the relationship between flow signal voids of the choriocapillaris and the structural and/ or decorrelation variables has not been well-evaluated to date.

In this study, we performed quantitative OCT and OCTA assessments in CSC patients after half-dose PDT. In addition, to understand the mechanism underlying the therapeutic effects of PDT, the relationship between OCT and OCTA variables was compared with the flow signal voids of the choriocapillaris simultaneously during the pre- and post-PDT periods.

## Materials and methods

### Patients

This retrospective case series included 91 consecutively treated naïve patients (age > 20 years) with chronic CSC (lasting longer than 6 months after onset of first symptom) without severe atrophy who received half-dose PDT for foveal subretinal serous detachment between January

1, 2016 and July 31, 2018. The study was conducted at the Ophthalmology Department of Veterans Health Service Medical Center and Seoul Shinsegae Eye Center. The diagnosis was based on the patient's symptom, visual acuity, fundoscopic examination, fluorescein angiography, indocyanine green angiography, Enhanced depth imaging (EDI)-OCT (Spectralis, Heidelberg Engineering, Heidelberg, Germany), and OCTA (Spectralis, Heidelberg Engineering, Heidelberg, Germany). Subjects were excluded if their symptoms lasted fewer than six months; exhibited any sign of choroidal neovascularization with fluorescein angiography, indocyanine green angiography or OCTA; previously underwent PDT, focal laser photocoagulation, or anti-VEGF treatment; had evidence of choroidal atrophy; had high myopia defined as a refractive error (spherical equivalent) $< -6.0$ diopters or an axial length $> 26.5$ mm; underwent continuous corticosteroid therapy; or had intraocular surgeries, poor signal strength, or severe artifacts due to saccadic eye movement. Ultimately, 56 eyes of 56 patients (nine patients received PDT in both eyes, and we randomly selected one eye in each of these patients) who had been followed up for a minimum of 6 months were enrolled in the study. Each patient underwent a comprehensive ophthalmological examination, including a review of medical and clinical history, measurement of best-corrected visual acuity (BCVA), slit-lamp biomicroscopy, intraocular pressure using Goldmann applanation tonometry, and refraction assessment. BCVA was measured using a standard Snellen unit chart, and the results were converted to logarithm of the minimal angle of resolution (logMAR) values for statistical analysis.

Approval was obtained from the ethics committee of Veterans Health Service Medical Center (2019-01-036-002). The protocol of this study adhered to the tenets of the Declaration of Helsinki.

## Half-dose photodynamic therapy

Patients satisfying the inclusion criteria underwent 4500–7300-μm sized half-dose PDT (Vitra PDT; Quantel Medical, Cournon-d'Auvergne, France) centered on the fovea. CSC was performed using half the standard dose (3 mg/m$^2$) of verteporfin (Visudyne; Novartis AG, Michigan, USA). An infusion of verteporfin was performed for 10 min, followed by 693-nm laser delivery 15 min after the start of infusion as a standard procedure (total light energy, 50 J/cm$^2$; 600 mW/cm$^2$ over 83 s).

## OCT image analysis

OCT images were generated using the horizontal SD-OCT cross-section (15 lines spaced 250 μm apart) over a thickness of nine subfields. For better image quality, 25–30 frames were averaged for each B-scan. The quality criteria included an automatic real-time score of 16, and a signal-to-noise ratio of 15 dB or higher, as described in a previous study [17]. Central subfield thickness (CST) was automatically measured with the inbuilt software and choroidal thickness was measured between Bruch's membrane beneath the retinal pigment epithelium (RPE) and the chorioscleral interface using a built-in caliper tool. At each visit, all eyes underwent EDI-OCT with a 30˚ (approximately 9 mm) × 20˚ (approximately 6 mm) and EDI-OCTA with a 15˚ (approximately 4.5 mm) × 10˚ (approximately 3.0 mm) macular 3D cube acquisition and five frames were averaged for each B-scan. Automated segmentation with manual correction was performed using the built-in software to generate en face reconstruction images of the choriocapillary layer.

Choroidal layer analysis of the OCT images was manually performed based on the methods described by Branchini et al. [18]. We selected 100 μm as the cut-off size of the large choroidal vessels in the horizontal OCT images through the fovea. Large choroidal vessels horizontally measuring 100 μm or more and located closest to the center of the fovea were selected by the

caliper function of SD-OCT. Three measurements of the thickness of the choroid were made. The first was the overall SFCT, which was defined as the distance between the hyperreflective line corresponding to Bruch's membrane beneath the RPE and the inner surface of the sclera. The second measurement was the subfoveal choroidal large vessel layer thickness (SFLVT). For this measurement, a horizontal line was drawn from the innermost point of the large choroidal vessels that intersected the SFCT measurement line perpendicularly. The thickness of the large choroidal vessel layer was defined as the distance from the point of intersection on the SFCT line to the inner surface of the sclera. The third measurement was the subfoveal choriocapillaris layer thickness (SFCCT). The SFCCT was obtained by subtracting the thickness of the large choroidal vessel layer from the overall SFCT. Two trained independent observers (H.S.Y and J.S.H) masked to the patients' clinical information measured choriocapillaris and choroidal thickness, and the means of these values were used for analysis. The thickness of the large choroidal vessel layer was derived by subtracting the thickness of the choriocapillaris layer from the SFCT.

## OCTA image analysis

The choriocapillary layer was visualized with the user setting mode that yields a 40-μm slab between 0 and 40 μm beneath Bruch's membrane. To minimize the projection effect of the other layers, the projection artifact removal mode of the built-in software (HEYEX, Heidelberg Eye Explorer 1.10.4, Heidelberg, Germany) was activated. Each CC OCT angiogram was exported from the Spectralis OCTA software and imported into the open-source Fiji software without a reference line using the F8 key (an expanded version of ImageJ; fiji.sc). Using a derivative of the method originally described by Spaide, each 13.5-mm$^2$ (4.5 mm × 3 mm) 8-bit image was binarized into black and white pixels using the software Fiji and the command path Image > Adjust > Auto Local Threshold with the Phansalkar method (Fig 1; radius = 15 pixels; uncheck the box labeled white objects on black background). Thresholded areas ≥ 1 white pixel were considered flow signal voids for the quantification of areas of decreased choriocapillary perfusion. The total number, average individual area, and total area of flow signal voids in the thresholded images were analyzed and extracted using the command path Analyze > Analyze Particles in Fiji.

## Statistical analyses

Statistical Product and Service Solution (SPSS; v21, IBM Corp. Armonk, NY, USA) was used for statistical analysis, and p values less than 0.05 were considered statistically significant. All

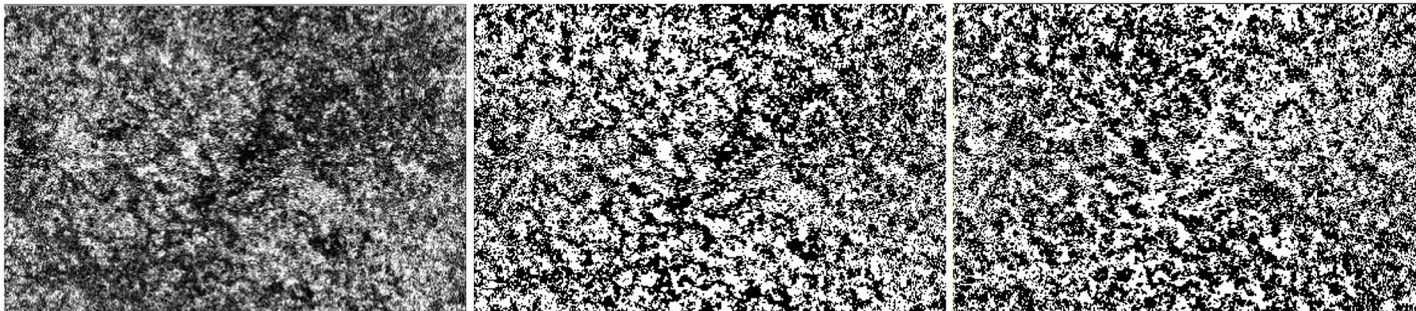

**Fig 1.** A reconstructed raw image of the choriocapillaris layer (A) and thresholding procedure using Image J. (B) The thresholded image of the raw image obtained using the Phansalkar method with white objects on a black background. (C) The thresholded image a of raw image using the Phansalkar method with inversion of grayscale.

data are presented as the mean ± standard deviation. When comparing two groups, either the independent t-test or the chi-square test was used depending on the number of samples and whether the assumption of a normal distribution was met. The normal distributions of the baseline characteristics of patients and the parameters of OCT and OCTA were assessed using a histogram and the Kolmogorov–Smirnov test.

Baseline characteristics were compared between baseline and 6 months after PDT by a paired T-test. A paired T-test was used to compare the quantitative variables of the OCT and OCTA between time points: at baseline and at 1, 3, and 6 months after PDT. Univariate and multivariate linear regression analyses were performed to ascertain the relationship between the mean total area of flow signal voids and the clinical data or the variables obtained using OCT and OCTA in CSC patients. To account for the effect of variables such as the elapsed time since PDT, age, intraocular pressure, refraction, and BCVA, the most appropriate multivariate model was selected through a stepwise linear regression method to model the relationship between variables. Multivariate stepwise linear regression analysis was performed without SFCT because of the high multicollinearity between the SFCT and SFCCT and/or SFLVT. The intra-class correlation coefficient (ICC) was used to examine interobserver agreement of the choriocapillaris and choroidal thickness measurements.

The correlation was classified as excellent if the ICC was greater than or equal to 0.6, fair if it was between 0.6 and 0.3, weak if it was between 0.3 and 0.1, and poor if it was 0.1 or below.

## Results

### Baseline characteristics

The average age of the patients was $47.8 \pm 6.4$ years. The patients exhibited symptoms such as decreased vision, metamorphopsia, and micropsia with an average pretreatment symptom duration of $9.2 \pm 1.6$ months. Forty-one patients were men (73.2%) and 15 were women (26.8%). All patients in this study exhibited complete resolution of SRF; 54 eyes (96.4%) showed SRF resolution at 1 month after PDT, and 56 eyes (100%) showed SRF resolution at 3 months after PDT. At baseline, the average BCVA (logMAR) was $0.45 \pm 0.23$, and the average intraocular pressure (IOP) was $13.9 \pm 2.4$ mmHg. The BCVA significantly improved, but IOP and refraction remained unchanged between baseline and at 6 months after PDT (Table 1). No significant complication was found after half-dose PDT according to follow-up assessments.

### Quantitative changes in the retina and choroid using OCT and OCTA

Table 2 shows quantitative measurements of OCT and OCTA parameters according to the time elapsed since the half-dose PDT. The CST, SFCT, SFCCT, SFLVT, mean number of flow

**Table 1. Baseline characteristics of the patients.**

| N = 56 | Baseline | At 6 months | p value [a] (paired T-test) |
|---|---|---|---|
| **Sex** | Male = 41 /Female = 15 | | |
| **Age** | 47.8 ± 6.4 (32–59) | | |
| **IOP** | 13.9 ± 2.4 | 13.8 ± 2.6 | 0.859 |
| **Refraction** | -0.368 ± 1.482 | -0.631 ± 1.386 | 0.219 |
| **BCVA (logMAR)** | 0.45 ± 0.23 | 0.16 ± 0.16 | <0.001 |

[a]paired T-test

BCVA = best corrected visual acuity; IOP = intraocular pressure

**Table 2. Quantitative values of optical coherence tomography and optical coherence tomography angiography variables according to the time elapsed after half-dose photodynamic therapy.**

| characteristics | Baseline | 4 wk | Pᵃ (baseline vs 4 wk) | 12 wk | Pᵃ (4 wk vs 3 mo) | 24 wk | Pᵃ (12 wks vs 6 mo) |
|---|---|---|---|---|---|---|---|
| **Central subfield thickness, μm** | 377.3±88.1 | 321.2 ±71.9 | <0.001 | 296.4±51.2 | 0.004 | 286.3±41.2 | 0.009 |
| **Subfoveal choroidal thickness, μm** | 460.1±96.0 | 408.6±81.9 | <0.001 | 391.1±96.1 | 0.089 | 375.0 ±84.8 | 0.098 |
| **Subfoveal choriocapillaris layer thickness, μm** | 49.9±12.2 | 51.7±9.7 | 0.16 | 52.3 ± 11.9 | 0.714 | 52.5±8.3 | 0.908 |
| **Subfoveal choroidal large vessel layer thickness, μm** | 410.2±87.5 | 356.9±81.8 | <0.001 | 338.8±100.4 | 0.084 | 332.5±85.7 | 0.103 |
| **Mean number of flow signal voids, n** | 502.5±323.0 | 688.5±274.1 | <0.001 | 903.0±273.6 | <0.001 | 947.8±306.6 | <0.001 |
| **Mean average size of flow signal voids, pixel²** | 204.2±157.8 | 91.4±45.1 | <0.001 | 65.5±32.1 | <0.001 | 58.2±26.7 | <0.001 |
| **Mean total area of flow signal voids, pixel²** | 67005.5 ±15645.1 | 54539.6 ±15541.0 | <0.001 | 53798.3 ±15834.7 | 0.392 | 49833.3 ± 15041.9 | 0.200 |

ᵃpaired T-test

BCVA = best corrected visual acuity; IOP = intraocular pressure

signal voids, mean average size of flow signal voids and mean total area of flow signal voids were 377.3 μm, 460.1 μm, 49.9 μm, 410.2 μm, 505.5, 204.2 pixel², and 67005.5 pixel², respectively. Fig 2 also shows that PDT induced SRF absorption, choroidal thinning and increased choroidal perfusion. The CST, SFCT, SFLVT, and mean total area of flow signal voids tended to decrease during follow-up (p<0.001), but the SFCCT did not change significantly (Fig 3;

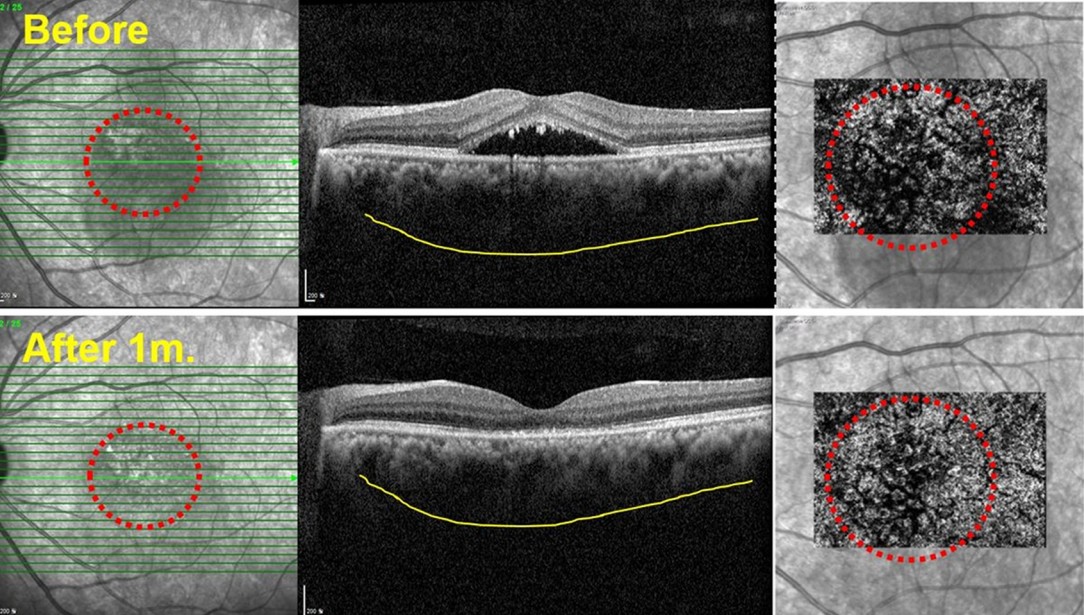

**Fig 2. Representative images from a 48-year-old man with central serous chorioretinopathy before and 1 month after half-dose photodynamic therapy.** (A) An optical coherence tomography (OCT) image obtained pre-treatment shows a dark area in the infrared image (red dotted circle, left) and thick choroidal thickness (yellow line). A reconstructed optical coherence tomography angiography (OCTA) image shows diffuse flow signal voids (red dotted circle, right) (B) An OCT image obtained post-treatment shows decreased dark area in the infrared image (red dotted circle, left) and decreased choroidal thickness (yellow line). A reconstructed OCTA image shows decreased flow signal voids (red dotted circle, right).

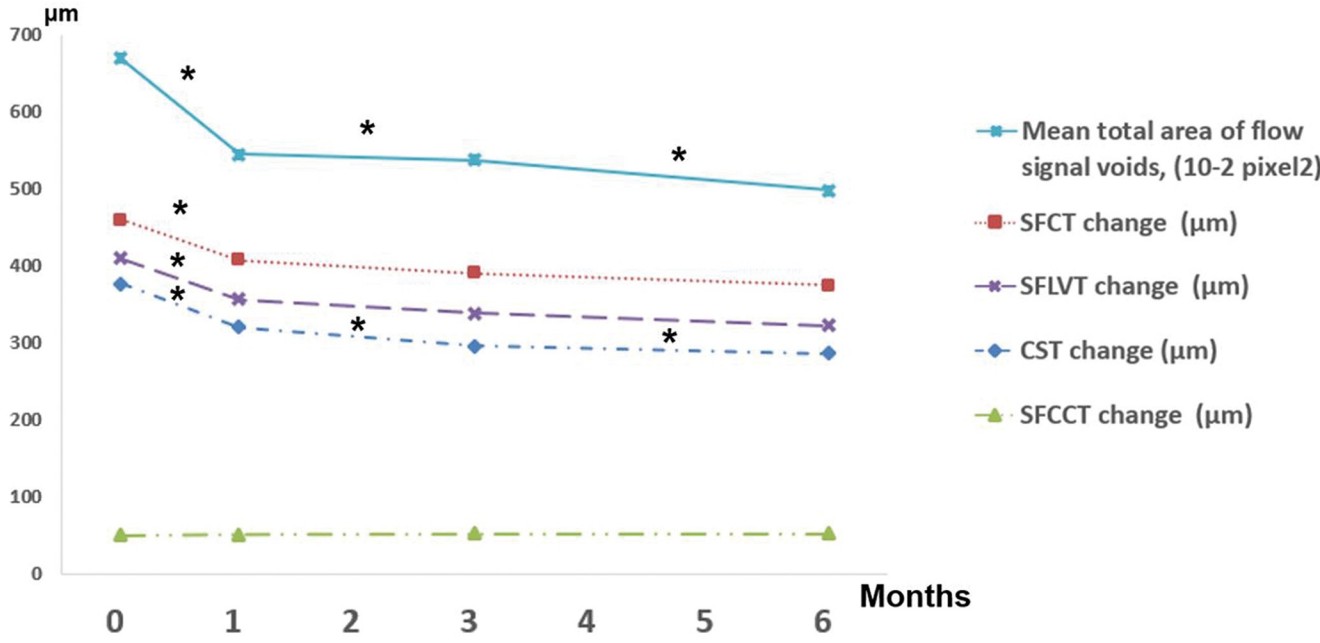

**Fig 3. Graph of the optical coherence tomography and angiography parameters before and after photodynamic therapy (PDT).** Mean total area of flow signal voids, subfoveal choroidal thickness (SFCT), subfoveal choroidal large vessel layer thickness (SFLVT), and central subfield thickness (CST) tended to decrease after PDT. However, subfoveal choriocapillaris layer thickness (SFCCT) exhibited a slight, statistically insignificant increase (all p ≥ 0.16). Asterisks (*) above the lines represent statistical significance (p < 0.05) between the two periods.

$p \geq 0.16$). Interobserver reproducibility of the SFCT and SFCCT were 0.943 (95% confidence interval = 0.913–0.970) and 0.893 (95% confidence interval = 0.860–0.919), respectively.

## The relationship between mean total area of flow signal voids and clinical data, CST, SFCT, SFCCT, or SFLVT

The CST and the mean total area of flow signal voids at baseline showed a moderate positive correlation (R = 0.468). Fig 4 shows the relationships between OCT variables and the mean total area of flow signal voids in all eyes at baseline and 1, 3, and 6 months after PDT (n = 226 eyes). CST exhibited weak positive correlation with the mean total area of flow signal voids (R = 0.198). SFCT and SFLVT was positively correlated with the mean total area of flow signal voids (R = 0.624 and 0.668, respectively). However, there was a weak negative correlation between SFCCT and the mean total area of flow signal voids (R = 0.190). To minimize the effect of SRF on flow signal void attenuation, we also used OCTA to analyze the relationship between OCT variables and the mean total area of flow signal voids in eyes without SRF at 1 month (n = 54 eyes), 3 months (n = 56 eyes), and 6 months (n = 56 eyes) after PDT (Fig 5). There was a slight negative correlation between the CST and the mean total area of flow signal voids (R = 0.223). SFCT and SFLVT were positive correlated with the mean total area of flow signal voids (R = 0.582 and 0.594 respectively). However, there was a weak negative correlation between SFCCT and the mean total area of flow signal voids (R = 0.215).

Table 3 shows the results of the univariate and multivariate analyses for determining the mean total area of flow signal voids. In the univariate analysis, time elapsed since PDT, BCVA, CST, SFCT, and SFLVT were significantly associated with the mean total area of flow signal

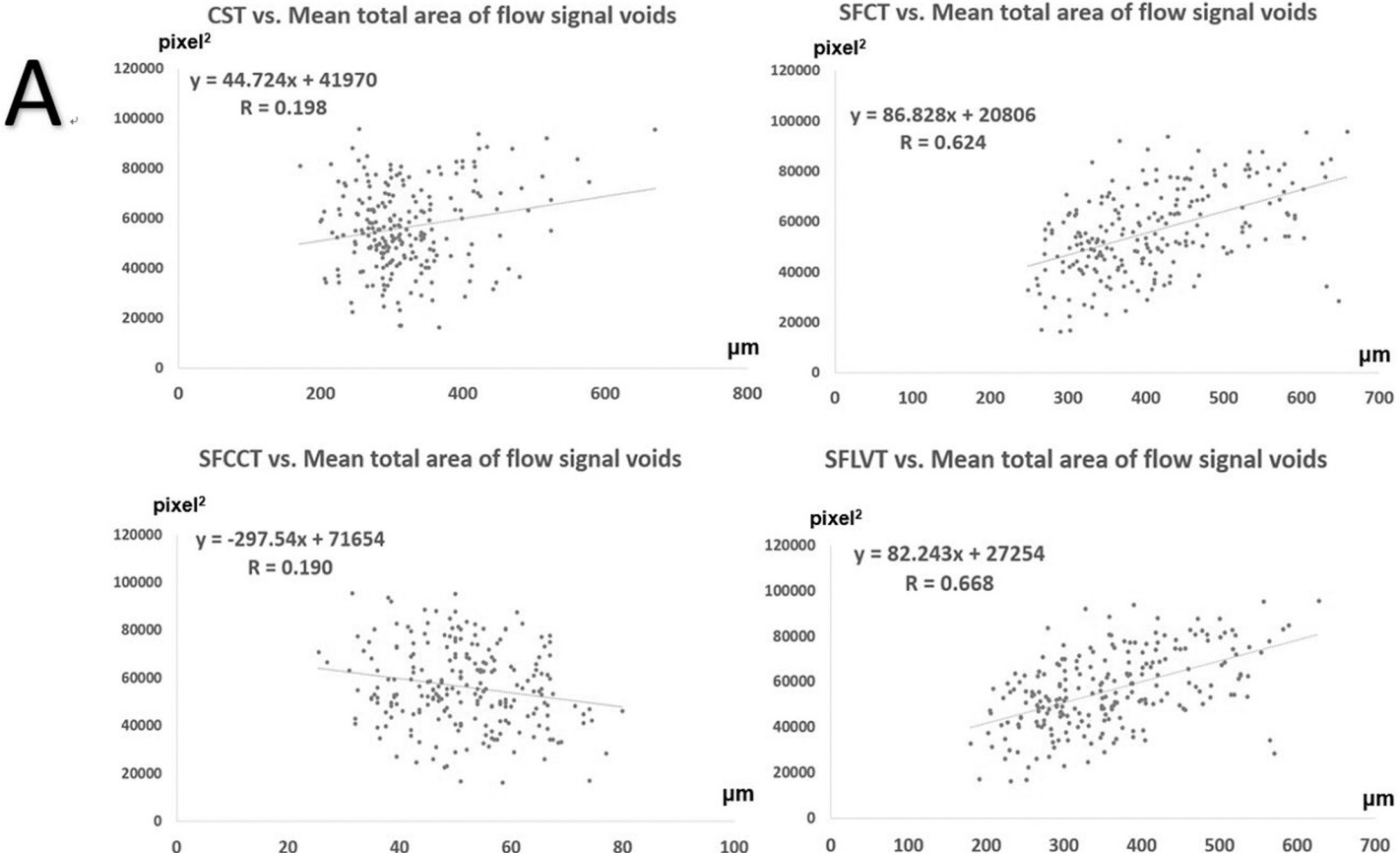

**Fig 4. Scatterplots of correlation and generalized equations between the mean total area of flow signal voids and central subfield thickness (CST), subfoveal choroidal thickness (SFCT), subfoveal choriocapillaris layer thickness (SFCCT), or subfoveal choroidal large vessel layer thickness (SFLVT), prior to and 6 months after photodynamic therapy in all eyes with central serous chorioretinopathy (n = 224).**

voids (p < 0.001, 0.011, <0.001, <0.001, and <0.001, respectively). In the multivariate analysis, CST and SFLVT showed positive linear correlations with the mean total area of flow signal voids (p < 0.001). SFCCT showed a negative linear correlation with the mean total area of flow signal voids (p = 0.013).

## Discussion

This study demonstrated that PDT leads to improved CST and BCVA in chronic CSC patients, as reported previously [2,19–21]. In addition, PDT can induce significant structural changes and alter choroidal blood flow. We also attempted to elucidate the pathophysiology of CSC and present a potential explanation for the mechanisms underlying the effects of PDT by using our findings of the structural and decorrelation parameters. Most importantly, we demonstrated a possible relationship between the thickness of the choriocapillaris determined by OCT and the choriocapillaris perfusion, which was identified as flow signal voids under OCTA.

There are marked changes to the ophthalmological structure and function following PDT, as demonstrated by OCT and OCTA. PDT induced SRF absorption and reduction of SFCT primarily by inducing shrinkage of the subfoveal choroidal large vessel layer (Fig 2). In addition, the decreased flow signal voids indicated improved choriocapillaris perfusion pressure.

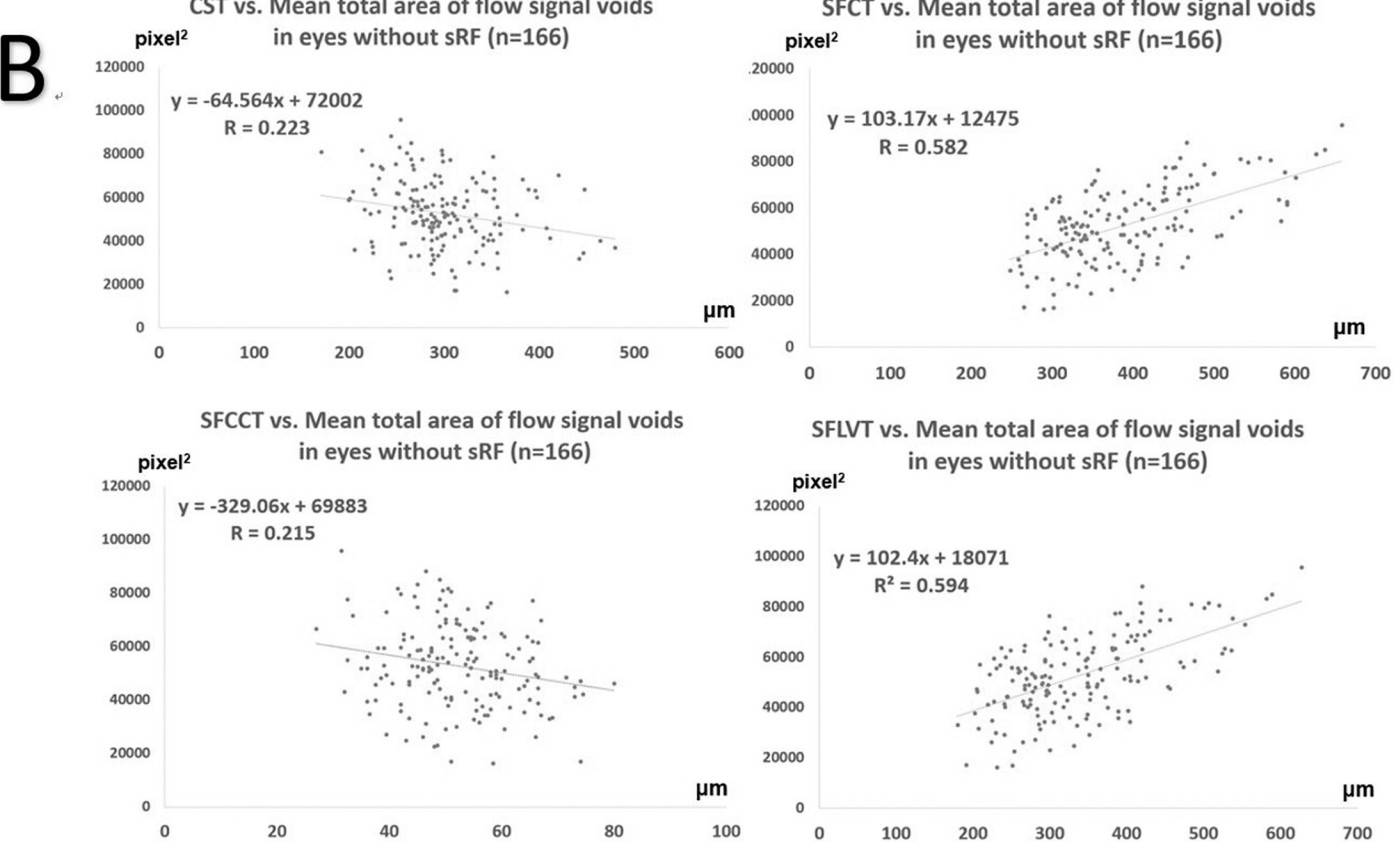

**Fig 5. Scatterplots of correlation and generalized equations between the mean total area of flow signal voids and central subfield thickness (CST), subfoveal choroidal thickness (SFCT), subfoveal choriocapillaris layer thickness (SFCCT), or subfoveal choroidal large vessel layer thickness (SFLVT) prior to and 6 months after photodynamic therapy in eyes with central serous chorioretinopathy, without subretinal fluid (SRF), after photodynamic therapy (n = 166).**

Table 2 and Fig 3 outline our findings of an abrupt decrease in CST, SFCT, and SFLVT, but not in SFCCT, which only exhibited a mild, statistically insignificant decrease. However, multivariate analysis showed a significant correlation between the mean total area of flow signal voids and SFCCT. These results imply that PDT may induce shrinkage of choroidal large vessels and increase the thickness of the choriocapillaris layer, subsequently increasing the choroidal microvasculature flow recovery.

Fig 6 highlights results that support our hypothesis of the potential mechanism by which PDT aids the recovery of choriocapillaris perfusion and SRF absorption. One possible mechanism is that PDT induces diffuse large-vessel shrinkage (Fig 6A). The subsequent release of the compression effect due to the enlarged large vessels may decrease pressure around the choriocapillaris and increase perfusion. The total choroidal thickness is also decreased because the SFLVT is mainly affected rather than the SFCCT. Another possible mechanism is that PDT changed the inner pressure of the choroidal large vessels via shrinkage (Fig 6B). Diffuse shrinkage of SFLVT, including large vessels and associated stromal tissue, may cause increased inner pressure in large vessels and induce increased choriocapillaris perfusion pressure as a result.

This study was based on the structural data obtained using OCT and the flow-related decorrelation data from OCTA. The higher axial resolution enabled the evaluation of both choroidal layer thickness and vasculature in healthy eyes [18,22,23]. However, the decreased flow signal voids appear to be related to the SRF (right red dotted circle, Fig 2) even though

**Table 3. Univariate and multivariate stepwise linear regression analysis for determining the mean total area of flow signal voids in patients with central serous chorioretinopathy before and after half-dose photodynamic therapy.**

| Factor | Univariate analysis | | Multivariate analysis (corrected R = 0.587) | |
|---|---|---|---|---|
| | Regression coefficient B | p-value | Standardized coefficient Beta | p-value |
| Elapsed time since PDT (before, at 1, 3, and 6 months post-PDT) | -0.286 | <0.001 | | |
| Age, years | -0.047 | 0.624 | | |
| Sex | 0.045 | 0.501 | | |
| IOP, mmHg | 0.006 | 0.953 | | |
| Refraction, Diopter | -0.101 | 0.287 | | |
| BCVA, logMAR | 0.238 | 0.011 | | |
| Central subfield thickness, μm | 0.290 | <0.001 | 0.363 | <0.001 |
| Subfoveal choroidal thickness, μm | 0.282 | <0.001 | | |
| Subfoveal choriocapillaris layer thickness, μm | -0.056 | 0.406 | -0.203 | 0.013 |
| Subfoveal choroidal large vessel layer thickness, μm | 0.290 | <0.001 | 0.400 | <0.001 |

BCVA = best-corrected visual acuity; IOP = intraocular pressure

Multivariate stepwise linear regression analysis was conducted without subfoveal choroidal thickness due to its high multicollinearity with subfoveal choriocapillaris layer thickness and subfoveal choroidal large vessel layer thickness. The regression equation using non-standardized B coefficients: $Y = 23557.929 + 77.413X_{1(\text{central subfield thickness})} - 339.178X_{2(\text{subfoveal choriocapillaris layer thickness})} + 72.458X_{3(\text{subfoveal large vessel layer thickness})}$

improved flow signal voids were also observed in the area surrounding the SRF. In relation to this finding, many studies have reported about projection artifacts [24–29]. Rochepeau et al. also reported the potential OCTA decorrelation signal attenuation and false-positive flow

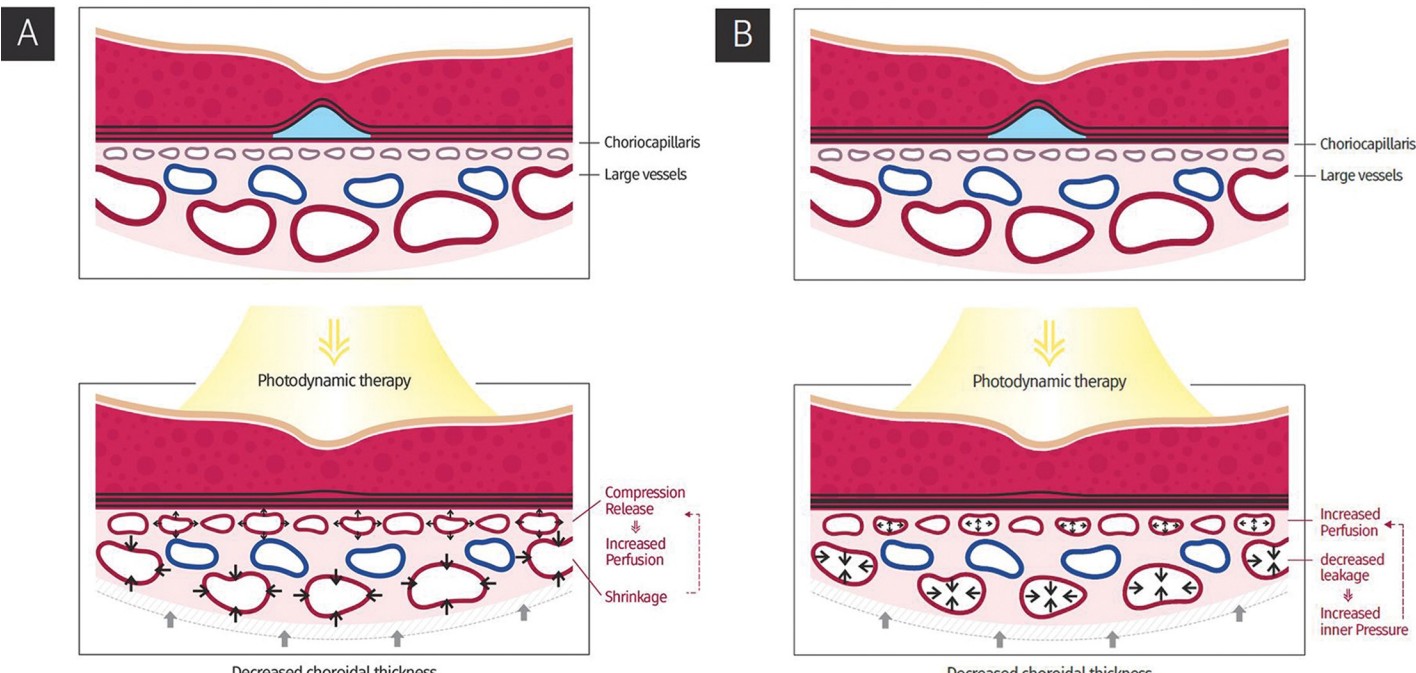

**Fig 6. The proposed mechanism for choroidal layer changes after photodynamic therapy (PDT).** (A) After PDT, choroidal large vessels continue to shrink, leading to decreased compression of the choriocapillaris by the large vessel layer. As a result, there is increased perfusion of the choriocapillaris and the total choroidal thickness is reduced after PDT. (B) After PDT, shrinkage of the large choroidal vessels may increase intravascular pressure as well as perfusion of the choriocapillaris. The total choroid thins as a result of the shrinkage of the large choroidal vessels.

impairment caused by SRF in CSC patients [30]; they have thus excluded all eyes with SRF. In the present study however, without this data, we could not analyze flow signal voids of the choriocapillaris in eyes with active CSC. In our study, CST appeared to show a positive relationship to the dark area of the choriocapillaris layer and affect the measurement of flow signal void areas that overlap with the hypo-signal area in the infrared image (Fig 2; in this line, for CC assessment with OCTA, it is important to evaluate not only the face flow image but also the structural image to exclude potential signal attenuation artifacts [22,23]). Thus, to minimize the potential signal attenuation artifacts caused by SRF, we also separately evaluated the correlation coefficient between CST and the mean total area of flow signal voids in eyes with and without SRF. Interestingly, there was a moderate positive correlation between CST and the mean total area of flow signal voids in eyes with fluid at baseline (regression coefficient B = 83.106, R = 0.468), but only a weak negative correlation between CST and the mean total area of flow signal voids in eyes without SRF after PDT (regression coefficient B = -64.564, R = 0.223). Thus, SRF can attenuate flow signal voids, but a thick CST without SRF may be associated with a well-perfused choriocapillaris with fewer flow signal voids. The relationship between the mean total area of flow signal voids and SFCT, SFCCT, and SFLVT was not largely influenced by the presence of SRF. Therefore, as Spaide [16] described, a reproducible quantitative analysis of the choriocapillaris was generally possible using flow signal voids in CSC patients with or without SRFs.

To minimize the decorrelation signal artifacts described above, we also checked the automated segmented layer and conducted manual correction if required. For simultaneous evaluation of the clinical variables, OCT variables, OCTA variables, and the effect of SRF on the signal in flow signal void attenuation, multivariate analysis of the relationship between mean total area of flow signal voids and other variables was performed (Table 3). The flow signal voids significantly affected the CST (p < 0.001), the SFCCT (p = 0.013), and SFLVT (p < 0.001) after adjusting for the effects of elapsed time since PDT, age, sex, IOP, refraction, and BCVA.

As described above, many reports have described increased choriocapillaris perfusion after PDT, as assessed using OCTA [13–15]. Cennamo et al. also reported vascular remodeling of the choriocapillaris after low-fluence PDT by showing a "narrow mesh" pattern of the choriocapillaris using OCTA in the SRF-absorbing group; however, the non-responder group showed persistence of abnormal vessels and a heterogeneous flow pattern rather than a "narrow mesh" pattern after PDT [31]. In addition, in a structural study using OCT, Izumi et al. reported that half-dose PDT reduces SFCT and alters the intrachoroidal structure [12]. In this study, the mean thickness of SFLVT decreased from 368.7 ± 112.7 μm at baseline to 292.2 ± 118.0 μm at 3 months (p < 0.0001), but the mean choriocapillaris-medium choroidal vessel layer thickness did not change from 44.2 ± 29.2 μm at baseline to 47.8 ± 25.7 μm at 3 months (p = 0.85). Our study also demonstrated increased perfusion of the choriocapillaris but found no significant changes in the thickness of the choriocapillaris during follow-up assessments. This disparity between vascular perfusion and thickness appears counterintuitive. However, multivariate analysis showed a negative relationship (p = 0.013) between the SFCCT and the mean total area of flow signal voids. Thus, the tiny change in SFCCT appeared statistically insignificant in the paired T-test, but the removal of confounding factors and direct analysis enabled us to identify the significant relationship between choroidal structure changes and vascular perfusion change, unlike previous studies.

One of the strengths of this study was the inclusion of patients who were carefully selected according to the strict criteria of our retina clinic. In addition, the present study only focused on treatment-naïve chronic CSC patients, and all analyzed eyes were of patients who finished 6 months of follow-up. The number of patients was sufficient for analysis of the relationships

among the clinical features, OCT parameters, and flow signal voids in OCTA. Despite these strengths, the present study had some limitations, including its retrospective design, which contributed to a lack of control groups. In addition, to fulfill the inclusion criteria, 35 patients were excluded (38.5%) and therefore selection bias may have been present. There are also major concerns regarding the shadowing effect of SRF, photoreceptor layer disruption, and RPE thickening and thinning. In the previous study [32], the difference in compensation was less than 10% of the total amount corrected when using flow signal voids with the Phansalkar method of reducing the shadowing artifact from superimposed retinal structure. Spectralis provides an EDI mode of OCTA while HEYEX provides the projection artifact removal mode for enhanced analysis of the outer retina and choroidal structures. Even when using these methods, the shadowing effect of the subretinal fluid and RPE attenuation cannot be fully controlled by image modulations. One of the best ways to control for compounding factors (retinal thickness for subretinal fluid and RPE thinning, BCVA-related to the integrity of the photoreceptor and RPE layer, choroidal thickness, age etc.) is statistical correction using multivariate analysis with related variables (flow signal voids, retinal thickness, age, sex, BCVA, subfoveal choriocapillaris layer thickness, etc.). In addition, we supplied all data with or without SRF and conducted subgroup analysis to determine the effect of SRF. Furthermore, the increased CST was found to be related to the decreased flow signal void in all data whereas decreased CST was related to the increased flow signal void. The flow signal void decrease was associated with CST thinning, but was unaffected by photoreceptor and RPE thinning. This indicates that the flow signal is mainly affected by not the image artifact but the chronic CSC itself. However, further validation of the statistical correction of compounding factors is required.

To the best of our knowledge, there have been no studies that 1) simultaneously measured choroidal thickness parameters using OCT and the mean total area of flow signal voids of the choriocapillaris using OCTA, 2) evaluated the direct effect of the SRF via quantitative measurements of OCT and OCTA parameters using correlation analysis, and 3) adjusted for the possible direct effect of SRF on the quantitative measurement and analysis of OCT and OCTA parameters by using flow signal void measurement methods and multivariate analysis to evaluate the effect of PDT on choroidal thickness thinning and increased choriocapillaris perfusion.

In conclusion, half-dose PDT induces absorption of SRF and increased BCVA in chronic CSC patients without significant complications. Half-dose PDT also induced thinning of the SFCT and increased choriocapillaris perfusion. In addition, SRF may affect the measurement of choriocapillaris perfusion when using the choriocapillaris flow signal void measurements, but statistically adjusting for the effects of SRF on choriocapillaris function revealed a close association between improved SFCCT and choriocapillaris recovery from hypoperfusion.

## Supporting information

**S1 File. Anonymized data of all patients.**
(XLSX)

## Author Contributions

**Conceptualization:** Hyun Seung Yang, Hyun Park, Kyung Sub Lee.

**Data curation:** Hyun Seung Yang, Tae Gu Kang, Ji Su Heo, Sangkyung Choi.

**Formal analysis:** Hyun Seung Yang.

**Funding acquisition:** Tae Gu Kang, Sangkyung Choi.

**Investigation:** Hyun Seung Yang, Jonghoon Park.

**Methodology:** Hyun Park, Ji Su Heo, Sangkyung Choi.

**Resources:** Ji Su Heo, Sangkyung Choi.

**Supervision:** Tae Gu Kang, Jonghoon Park, Kyung Sub Lee, Sangkyung Choi.

**Validation:** Hyun Seung Yang, Hyun Park.

**Visualization:** Hyun Seung Yang.

**Writing – original draft:** Hyun Seung Yang.

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
