## [Decision Letter · Decision Letter 0]

19 Sep 2019

PONE-D-19-24994

Quantitative evaluation of choriocapillaris using optical coherence tomography and optical coherence tomography angiography in patients with central serous chorioretinopathy after half-dose photodynamic therapy

PLOS ONE

Dear Dr. CHOI,

Thank you for submitting your manuscript to PLOS ONE. After careful consideration, we feel that it has merit but does not fully meet PLOS ONE’s publication criteria as it currently stands. Therefore, we invite you to submit a revised version of the manuscript that addresses the points raised during the review process.

We would appreciate receiving your revised manuscript by Nov 03 2019 11:59PM. To enhance the reproducibility of your results, we recommend that if applicable you deposit your laboratory protocols in protocols.io, where a protocol can be assigned its own identifier (DOI) such that it can be cited independently in the future. For instructions see: http://journals.plos.org/plosone/s/submission-guidelines#loc-laboratory-protocols

We look forward to receiving your revised manuscript.

Kind regards,

Ireneusz Grulkowski, PhD

Academic Editor

PLOS ONE

Journal Requirements:

NO: The funders had no role in study design, data collection and analysis, decision to publish, or preparation of the manuscript

a) Please provide an amended Funding Statement that declares *all* the funding or sources of support received during this specific study (whether external or internal to your organization) as detailed online in our guide for authors at http://journals.plos.org/plosone/s/submit-now.  

b) Please state what role the funders took in the study.  If any authors received a salary from any of your funders, please state which authors and which funder. If the funders had no role, please state: "The funders had no role in study design, data collection and analysis, decision to publish, or preparation of the manuscript."

Reviewers' comments:

Reviewer's Responses to Questions

**Comments to the Author**

1. Is the manuscript technically sound, and do the data support the conclusions?

Reviewer #1: Partly

Reviewer #2: Partly

2. Has the statistical analysis been performed appropriately and rigorously? 

Reviewer #1: Yes

Reviewer #2: Yes

3. Have the authors made all data underlying the findings in their manuscript fully available?

Reviewer #1: Yes

Reviewer #2: Yes

4. Is the manuscript presented in an intelligible fashion and written in standard English?

Reviewer #1: Yes

Reviewer #2: Yes

5. Review Comments to the Author

Reviewer #1: The authors evaluated choriocapillaris changes by evaluating flow voids and choroidal thickness in chronic CSC treated with half dose PDT and evaluated the patients at 1, 3 and 6 months after treatment.

It is not clear why the mean follow- up was 4.4months+/-1.8 months if all the patients were evaluated after 6 months.

The classification of chronic CSC is not quite clear. The authors included patients that after three months of CSC did not spontaneous resolution? Is it correct? Usually 6 months are considered a chronic CSC. There is no control group in this study. This is a significant limitation of this retrospective study.

In the introduction, please update the available treatments for CSC with subthreshold micropulse laser.

In the methods section, the choriocapillaris flow voids evaluation should be better explained. Did the authors use the method proposed by Zhang et al (IOVS 2018)? It should be better explained how the artifacts due to the presence of fluid were corrected?

Reviewer #2: Seung Yang et al. present a retrospective analysis of changes in the choriocapillaris and choroid following PDT in patients with CSC by means of OCT and OCT-A. Though the idea behind this study is very interesting and relevant, there are several issues needing clarification. Please find a more detailed description in the comments below.

1. Abstract: Results: the number of patients/ eyes is missing and should be mentioned here.

It is not clear, for what time interval the results and p-values are presented – for 1, 3 or 6 months?

2. Definition of chronic CSC: The authors define “chronic” CSC as CSC with symptoms >3months. The common definition of “chronic”, however, is a duration of symptoms/ disease >6 months.

3. The illustration of in- and exclusion criteria is confusing, a better structure is needed here.

4. Were OCT-A - CC images averaged? This may be crucial for further (quantitative) analysis.

5. Details on OCT-A image acquisition are missing: were these 3mm x 3mm images (the authors mention a 9mm² scan)? How many scans were used? Were the scans centered on the fovea? And if so, in a quite small test field of 9mm² , in some cases the “area of interest” was probably not (completely) included within these scans – how did the authors ensure that this was the case?

Why did the authors solely analyze the choriocapillaris and not the choroid (e.g. as inner, mid and outer choroid sections)?

6. Statistics: t-test: How do the authors presume their data to be normally distributed?

7. The measurement of IOP is mentioned in the results section, but not in the methods section.

8. Figure 2: This is in the reviewer’s opinion the most important issue in this manuscript: How can the authors exclude that the observed phenomenon (decrease of flow voids in OCT-A) is caused by a “shadowing” artifact that is induced by superimposing structures (e.g. subretinal fluid) and thus decreases when the fluid diminishes? Have the authors thoroughly looked at all retinal layers and the ORCC-layer for possible causes of shadowing artifacts and morphological changes between pre- and posttreatment (in the OCT-A images in Fig 2-A and B, there is for instance a visible shadowing artifact from superimposing retinal vessels)? Have the authors applied corrections for projection artifacts? Could the presence of neovascular membranes be definitely excluded? Quaranta-El Maftouhi et al, (AJO, 2015) describe small hyperreflective undulations of the RPE in OCT-scans of patients with chronic CSC, that were shown to represent neovascular complexes. In the OCT-scan shown in figure 2-B, we can also see such small hyperreflective undulations of the RPE…

Further, the OCT B-scan in Figure 2-A shows shadowing artifacts at the level of the RPE/ CC induced by hyperreflective structures at the level of the elevated outer retina, while this is not visible in Figure 2-B.

Further, Fig. 2 A and B do not show the same B-scans (scan 12/25 in Figure 2-A and 13/25 in Figure 2-B).

The OCT-A images in Figure 2-A and -B do not show typical morphological features of the CC-layer, but rather look like mid-choroid sections (see for example Wang JC, TVST, 2018). Which OCT-A layers were used here?

The exact size and position of the OCT-A en face image should be indicated in the IR-image.

6. PLOS authors have the option to publish the peer review history of their article (what does this mean?). If published, this will include your full peer review and any attached files.

Reviewer #1: No

Reviewer #2: No

---

## [Author Response · Author response to Decision Letter 0]

24 Nov 2019

Related corrections were attached as files.

---

## [Decision Letter · Decision Letter 1]

11 Dec 2019

PONE-D-19-24994R1

Quantitative evaluation of choriocapillaris using optical coherence tomography and optical coherence tomography angiography in patients with central serous chorioretinopathy after half-dose photodynamic therapy

PLOS ONE

Dear Dr. CHOI,

Thank you for submitting your manuscript to PLOS ONE. After careful consideration, we feel that it has merit but does not fully meet PLOS ONE’s publication criteria as it currently stands. Therefore, we invite you to submit a revised version of the manuscript that addresses the points raised during the review process.

We would appreciate receiving your revised manuscript by Jan 25 2020 11:59PM. To enhance the reproducibility of your results, we recommend that if applicable you deposit your laboratory protocols in protocols.io, where a protocol can be assigned its own identifier (DOI) such that it can be cited independently in the future. For instructions see: http://journals.plos.org/plosone/s/submission-guidelines#loc-laboratory-protocols

We look forward to receiving your revised manuscript.

Kind regards,

Ireneusz Grulkowski, PhD

Academic Editor

PLOS ONE

Additional Editor Comments (if provided):

Please, make minor editing to the manuscript.

Reviewers' comments:

Reviewer's Responses to Questions

**Comments to the Author**

1. If the authors have adequately addressed your comments raised in a previous round of review and you feel that this manuscript is now acceptable for publication, you may indicate that here to bypass the “Comments to the Author” section, enter your conflict of interest statement in the “Confidential to Editor” section, and submit your "Accept" recommendation.

Reviewer #1: All comments have been addressed

Reviewer #2: (No Response)

2. Is the manuscript technically sound, and do the data support the conclusions?

Reviewer #1: Yes

Reviewer #2: Yes

3. Has the statistical analysis been performed appropriately and rigorously? 

Reviewer #1: Yes

Reviewer #2: N/A

4. Have the authors made all data underlying the findings in their manuscript fully available?

Reviewer #1: Yes

Reviewer #2: Yes

5. Is the manuscript presented in an intelligible fashion and written in standard English?

Reviewer #1: Yes

Reviewer #2: (No Response)

6. Review Comments to the Author

Reviewer #1: (No Response)

Reviewer #2: The authors did a good job in revising the manuscript, and addressed most of the issues raised by the reviewers. However, some minor issues remain to be resolved, and in terms of style and grammar, a linguistic revision would improve the readability of the manuscript.

1. Page 6: lines 94-97: choroidal neovascularizations are mentioned twice as exclusion criterion

2. Abstract: 56 eyes of 56 patients were considered for the analysis, but 91 patients were included in the case series – there is a discrepancy of case numbers in the main text (methods section) and the abstract (methods section). Suggest to include the numbers in the results section of the abstract (not in the methods section) for more clarity.

3. Answer to reviewer #2, question 8:

The authors state that “As you can observe in the figure 2, the decreased flow signal void is not confined to the area of SRF, but the outside of SRF area.”

The reviewer disagrees with this statement, as the field for OCT-A evaluation is obviously chosen too small, and does not appear to include an area without SRF.

7. PLOS authors have the option to publish the peer review history of their article (what does this mean?). If published, this will include your full peer review and any attached files.

Reviewer #1: No

Reviewer #2: No

---

## [Decision Letter · Decision Letter 2]

27 Dec 2019

Quantitative evaluation of choriocapillaris using optical coherence tomography and optical coherence tomography angiography in patients with central serous chorioretinopathy after half-dose photodynamic therapy

PONE-D-19-24994R2

Dear Dr. CHOI,

We are pleased to inform you that your manuscript has been judged scientifically suitable for publication and will be formally accepted for publication once it complies with all outstanding technical requirements.

With kind regards,

Ireneusz Grulkowski, PhD

Academic Editor

PLOS ONE

Additional Editor Comments (optional):

Reviewers' comments:

Reviewer's Responses to Questions

**Comments to the Author**

1. If the authors have adequately addressed your comments raised in a previous round of review and you feel that this manuscript is now acceptable for publication, you may indicate that here to bypass the “Comments to the Author” section, enter your conflict of interest statement in the “Confidential to Editor” section, and submit your "Accept" recommendation.

Reviewer #2: All comments have been addressed

2. Is the manuscript technically sound, and do the data support the conclusions?

Reviewer #2: (No Response)

3. Has the statistical analysis been performed appropriately and rigorously? 

Reviewer #2: (No Response)

4. Have the authors made all data underlying the findings in their manuscript fully available?

Reviewer #2: (No Response)

5. Is the manuscript presented in an intelligible fashion and written in standard English?

Reviewer #2: (No Response)

6. Review Comments to the Author

Reviewer #2: (No Response)

7. PLOS authors have the option to publish the peer review history of their article (what does this mean?). If published, this will include your full peer review and any attached files.

Reviewer #2: No

---

## [Editor Report · Acceptance letter]

6 Jan 2020

PONE-D-19-24994R2 

Quantitative evaluation of choriocapillaris using optical coherence tomography and optical coherence tomography angiography in patients with central serous chorioretinopathy after half-dose photodynamic therapy 

Dear Dr. Choi:

I am pleased to inform you that your manuscript has been deemed suitable for publication in PLOS ONE. Congratulations! Your manuscript is now with our production department. 

With kind regards,

on behalf of

Dr. Ireneusz Grulkowski 

Academic Editor

PLOS ONE